# Exploring the Role of Biomarkers Associated with Alveolar Damage and Dysfunction in Idiopathic Pulmonary Fibrosis—A Systematic Review

**DOI:** 10.3390/jpm13111607

**Published:** 2023-11-14

**Authors:** Alexandra-Simona Zamfir, Mihai Lucian Zabara, Raluca Ioana Arcana, Tudor Andrei Cernomaz, Andreea Zabara-Antal, Marius Traian Dragoș Marcu, Antigona Trofor, Carmen Lăcrămioara Zamfir, Radu Crișan-Dabija

**Affiliations:** 1Clinical Hospital of Pulmonary Diseases, 700115 Iasi, Romania; simona-zamfir@umfiasi.ro (A.-S.Z.); arcana.raluca@umfiasi.ro (R.I.A.); antigona.trofor@umfiasi.ro (A.T.); radu.dabija@umfiasi.ro (R.C.-D.); 2Department of Medical Sciences III, Faculty of Medicine, University of Medicine and Pharmacy “Grigore T. Popa”, 700115 Iasi, Romania; 3Department of Surgery, Faculty of Medicine, University of Medicine and Pharmacy “Grigore T. Popa”, 700115 Iasi, Romania; 4Clinic of Surgery (II), St. Spiridon Emergency Hospital, 700111 Iasi, Romania; 5Doctoral School of the Faculty of Medicine, University of Medicine and Pharmacy “Grigore T. Popa”, 700115 Iasi, Romania; 6Regional Institute of Oncology, 700483 Iasi, Romania; 7Department of Medical Sciences I, Faculty of Medicine, University of Medicine and Pharmacy “Grigore T. Popa”, 700115 Iasi, Romania; 8Department of Morpho-Functional Sciences I, Faculty of Medicine, University of Medicine and Pharmacy “Grigore T. Popa”, 700115 Iasi, Romania; carmen.zamfir@umfiasi.ro

**Keywords:** idiopathic pulmonary fibrosis, biomarkers, Krebs von den Lungen 6, surfactant proteins, MUC5B, αvβ integrins, tumor markers, Clara cell secretory protein, telomere shortening

## Abstract

Background: Idiopathic pulmonary fibrosis (IPF) is one of the most aggressive forms of interstitial lung diseases (ILDs), marked by an ongoing, chronic fibrotic process within the lung tissue. IPF leads to an irreversible deterioration of lung function, ultimately resulting in an increased mortality rate. Therefore, the focus has shifted towards the biomarkers that might contribute to the early diagnosis, risk assessment, prognosis, and tracking of the treatment progress, including those associated with epithelial injury. Methods: We conducted this review through a systematic search of the relevant literature using established databases such as PubMed, Scopus, and Web of Science. Selected articles were assessed, with data extracted and synthesized to provide an overview of the current understanding of the existing biomarkers for IPF. Results: Signs of epithelial cell damage hold promise as relevant biomarkers for IPF, consequently offering valuable support in its clinical care. Their global and standardized utilization remains limited due to a lack of comprehensive information of their implications in IPF. Conclusions: Recognizing the aggressive nature of IPF among interstitial lung diseases and its profound impact on lung function and mortality, the exploration of biomarkers becomes pivotal for early diagnosis, risk assessment, prognostic evaluation, and therapy monitoring.

## 1. Introduction

Interstitial lung diseases (ILDs) include a heterogenous group of more than 100 pulmonary disorders distinguished by extensive inflammation and fibrotic changes in the pulmonary tissues. The classification of ILDs involves the analysis of histopathological specimens and radiological criteria correlating with clinical factors. This process typically requires a comprehensive assessment due to their intricate characteristics [1,2,3].

Idiopathic pulmonary fibrosis (IPF) is one of the most aggressive forms of ILDs, marked by an ongoing, chronic fibrotic phenomena causing gradual deterioration of lung function, leading to respiratory failure and death [4,5,6]. IPF frequently affects men; smoking and a history of professional inorganic dust exposure are also risk factors [7].

Early IPF detection is particularly challenging as symptoms are initially mild and non-specific; a definite IPF diagnosis usually relies on high-resolution computed tomography of the chest (HRCT) and lung tissue biopsy with the additional burden of ruling out other morbid conditions that may cause pulmonary fibrosis [7,8,9,10].

With the advent of antifibrotic drugs, early IPF diagnosis became essential in order to preserve the quality of life; therefore, biomarkers usable for risk assessment, early detection, prognosis, or management of the disease became necessary and sought after [11]. Potential biomarkers might reflect various biological phenomena: alveolar epithelial cell damage and dysfunction, extracellular matrix remodeling, fibroblast proliferation, and immune dysfunction/inflammation [12,13].

A precise classification of these biomarkers is challenging since their exact functions in IPF remain incompletely understood, and ongoing studies are continually progressing in this area. However, biomarkers that proved to be of interest and helpful in managing IPF can be systematized in eight classes from a clinical/physiopathological point of view:Serum biomarkers, such as Krebs von den Lungen-6 (KL-6), surfactant protein D (SP-D), and serum matrix metalloproteinase 7 (MMP-7) have been shown to be useful in the diagnosis of IPF. However, their sensitivity and specificity are not high enough to be used as standalone diagnostic tools;Bronchoalveolar lavage fluid biomarkers, such as SP-D, KL-6, and MMP-7 have been found in bronchoalveolar lavage fluid samples from patients with IPF. Their diagnostic accuracy is limited by the fact that they are not specific to IPF and can be found in other lung diseases;Exhaled breath condensate biomarkers, such as hydrogen peroxide, nitric oxide, and leukotrienes have been studied in the context of IPF diagnosis. However, their diagnostic accuracy is limited by the fact that they are not specific to IPF and can be influenced by other factors such as smoking;Lung tissue biomarkers, similar to serum biomarkers, have been studied in the context of IPF diagnosis but their diagnostic accuracy is limited since confirming their presence in the lung tissue requires invasive procedures such as lung biopsy;Genetic biomarkers, such as mucin 5B (MUC5B) and telomerase reverse transcriptase (TERT) have been associated with an increased risk of developing IPF, but their diagnostic accuracy is limited by the fact that they are not specific to IPF and can also be found in healthy individuals;Proteomic biomarkers, such as serum amyloid (SAA) and α1 antitrypsin (AAT) seem to be useful in the IPF diagnosis, but their accuracy is limited by their non-specificity to IPF and can be found in other lung diseases;Metabolomic biomarkers, such as sphingolipids and glycerophospholipids have been evoked in the context of IPF diagnosis, but their values are highly biased by the influence of rather uncontrollable factors, such as diet;MicroRNA biomarkers, such as miR-21 and miR-155 seem to have a future in predicting and accurately diagnosing IPF, but their diagnostic accuracy is limited by the requirement of specialized detection equipment [14,15,16,17].

The understanding of IPF has shifted from fibrosis triggered by inflammation to irregular wound healing due to repetitive injuries to alveolar epithelial cells (AEC). These injuries accumulate in the alveolar epithelium, reaching a threshold that triggers dysfunctional activation of AECs. These activated cells then set off profibrotic signaling pathways involving growth factors and chemokines. This process leads to the development of fibroblastic clusters and excessive extracellular matrix deposition, resulting in a proliferative fibrotic condition, where fibroblastic foci become the primary sites of profibrotic responses [12].

We aim to review available data on biomarkers relevant to alveolar epithelial cell damage and assess their place in managing idiopathic pulmonary fibrosis.

## 2. Materials and Methods

This review was conducted through a detailed evaluation of existing articles on idiopathic pulmonary fibrosis. A systematic search of the relevant literature was performed using established databases such as PubMed, Scopus, and Web of Science. The search terms employed in this study were carefully chosen to capture the relevant literature on biomarkers associated with alveolar epithelial cell damage and dysfunction in IPF. We have included in this manuscript the specific search terms used, such as: “biomarkers”, “idiopathic pulmonary fibrosis”, “Krebs von den Lungen 6”, “surfactant proteins”, “MUC5B”, “tumor markers”, “Clara cell secretory protein”, “telomere shortening”, “cleaved cytokeratin 18”, and “alpha-v beta-1 (αvβ1) integrin” or “alpha-v beta-6 (αvβ6) integrin”. The inclusion and exclusion criteria were applied to select studies meeting the review’s objectives. The inclusion criteria encompassed studies directly investigating biomarkers in IPF, with a focus on those associated with alveolar damage and dysfunction, as well as peer-reviewed journals that discussed early diagnosis, risk assessment, prognostic evaluation, or treatment progress monitoring in IPF. The exclusion criteria involved studies primarily concentrating on biomarkers in other pulmonary conditions unrelated to IPF, studies lacking essential data for analysis, or those conducted solely on animal models. Selected articles were mainly assessed for relevance, with data extracted and synthesized to provide a clear overview of the current state of knowledge on the existing biomarkers in IPF. The PRISMA flow diagram illustrates the article selection process (Figure 1).

## 3. Results

### 3.1. Biomarkers

The advent of biomarkers (also known as biological markers) brought significant changes in clinical practice and the management of some diseases—the impact is evident for, at least, neoplastic disorders—such as digestive solid tumors or prostate cancer [18,19]. An ideal biomarker should be easily obtainable, consistently measurable, and suitable for repeated monitoring while offering several benefits compared to current medical assessments in terms of simplicity, time efficiency, or cost-effectiveness [20].

There is no such ideal biomarker yet identified for idiopathic pulmonary fibrosis, but there are compelling data suggesting that a combination of biomarkers might have a higher rate of success and accuracy in predicting the disease progression [6]. Several biomarkers identified in the peripheral blood exhaled breath condensate or bronchoalveolar lavage offer additional information and should complement imagistics or invasive procedures such as transbronchial or surgical lung biopsy [21].

The disruption of the alveolar basement membrane represents a critical feature of idiopathic pulmonary fibrosis (IPF), triggering abnormal activation of alveolar epithelial cells, increasing the migration of inflammatory and mesenchymal cells into the alveolar space; this process is accompanied by the aberrant cell senescence of the epithelial cells and fibroblasts of the alveoli [22]. Signs of epithelial cell damage hold promise as diagnostic and prognostic biomarkers for IPF, consequently offering valuable support in its clinical care [23]. The main biomarkers associated with alveolar epithelial cell dysfunction are represented by Krebs von den Lungen 6 (KL-6), surfactant proteins, mucin 5B (MUC5B), oncomarkers (CA 15-3, CA 125, CA 19-9, and CEA), Clara cell secretory protein (CC16), telomeres shortening, cleaved cytokeratin 18 (cCK-18), alpha-v beta-6 (αvβ6), and alpha-v beta-1 (αvβ1) integrin [13,22,24].

### 3.2. Krebs von den Lungen 6 (KL-6)

Alveolar type 2 epithelial cells (AEC-II) produce a mucinous glycoprotein named Krebs von den Lungen 6, well-known as MUC1, which contains three segments: an extracellular domain (MUC1-N), a singular transmembrane region, and a cytoplasmic tail. The KL-6 epitope domain, elevated in IPF, is included in the extracellular environment. The release of KL-6 following injury is correlated with lung fibrosis and epithelial-mesenchymal transition. While there is evidence associating serum KL-6 levels with IPF progression, a knowledge gap persists regarding the biological activation and cell signaling of the MUC1 cytoplasmic tail in IPF. During the onset of pulmonary fibrosis, the discharge of the extracellular KL6/MUC1 domain results from the proteolytic cleavage near the plasma membrane, mediated by heightened levels of metalloprotease. Fibroblast activity is modulated by the soluble KL6/MUC1, which activates still unidentified targets. Antibodies against KL6 have shown efficacy in alleviating pulmonary fibrosis induced by bleomycin [25].

When AEC-IIs are affected, KL-6 is released into the circulation, enabling it to be detected and measured in the serum [22,23,26]. Even though there are some countries, such as Japan, where KL-6 serves as a diagnostic biomarker for IPF, several studies revealed that this glycoprotein is more effective as a predictive biomarker [22,27].

In IPF, the evolution might be favorable if the initial value of KL-6 in the serum is below 1000 U/mL and there is no continuous rise rather than in situations with ongoing elevated KL-6 or values equal to or higher than 1000 U/mL at the outset [22].

Acute exacerbations of IPF might also be predicted by a high initial serum KL-6 value (more than 1300 U/mL) [28].

KL-6 initial serum level could also potentially serve as a significant means to evaluate the effectiveness of treatment with nintedanib, as its value tends to decrease during therapy [29,30].

### 3.3. Surfactant Proteins

Alveolar epithelial type II cells are also responsible for producing lung surfactant, a critical factor not only in preventing lung collapse during the process of breathing, but also in immunity, due to its components (lipids and proteins) and their specific properties [31,32,33,34].

AEC-II cells are accountable for the synthesis, secretion, and recycling of all surfactant components. This process reduces surface tension, enabling normal breathing at transpulmonary pressure. Biochemical studies of the surfactant reveal a composition of approximately 90% lipid and 10% protein. The inquiry arises since, even though AEC-II possesses the capacity for renewal and self-regulation, they still undergo chronic injury in IPF. A pertinent observation from familial IPF suggests that mutations in surfactant proteins, along with telomere mutations, are correlated with chronic AEC-II cell injury and apoptosis [35].

Two of the proteins linked to lung surfactant, SP-A and SP-D, possess significant roles in natural immunity [31,34]. In contrast, the other two proteins (SP-B and SP-C) exhibit strong hydrophobic properties and enhance the reduction in surface tension while interacting with the lipids from the surfactant [31].

Research on variations in SP-A and SP-D concerning polymorphisms and protein levels in bronchoalveolar lavage and blood has revealed a connection with various pulmonary conditions, including IPF [36].

SP-A, encoded by SFTPA1 and SFTPA2 genes, is produced by airway epithelial cells. Mutations in SFTPA are linked to the development of ILDs and susceptibility to pulmonary adenocarcinoma. SP-A, comprising a glycosylate C-terminal lectin domain and an NH2-terminal collagenous domain, forms trimeric complexes and multimers in the airway, essential to ensure the surfactant lipid particle structure. Certain SFTPA mutations lead to misfolded protein responses and epithelial injury. SP-A mutations activate TGF-β signaling, potentially playing a role in the development of alveolar lung disease [36].

Similar to SP-A, SP-D is formed by three polypeptide chains interconnected by disulfide bonds, and all polypeptide chains are replicas of the SFTD gene. The regulatory mechanism governing the transcription of SP-D is not well comprehended, and there is a limited number of studies compared to the extensive research on SP-A regulation.

Polymorphisms in SP-A1, particularly the 6A4 allele markers, have been proposed as potential risk factors for IPF. Conversely, no examined variants of SP-D showed correlations with IPF. The concentration of SP-D was additionally discovered to correlate with the degree of collapse of the pulmonary parenchyma and the annual rate of deterioration in lung function. SP-A and SP-D demonstrate the potential to stimulate the synthesis of collagen and metalloproteinase, suggesting a plausible direct interaction in the pathogenesis of IPF [37].

Alongside Krebs von den Lungen, SP-A and SP-D are valuable biomarkers in prognosis and monitoring therapy progress, as their values decrease after treatment with antifibrotic drugs [38,39].

### 3.4. The Mucin MUC5B

Mucus clearance represents the primary mechanical defense process within the lungs [40]. The airway mucus contains several glycosylated proteins of substantial size (up to three million Daltons per monomer), called mucins, which ensure its gel-like structure [40,41,42]. Even though various genes are identified, it is essential to state that only 11 of them are localized in the lungs, and among these, only two exhibit high levels of expression: MUC5B and MUC5AC [42].

In IPF, MUC5B seems to have predictive value and can also be used for prognosis [43]. Previous studies showed that the functionality of the MUC5B promoter in IPF is enhanced by the existence of its rs35705950 variant, found in 38% of the patients suffering from IPF [44,45].

The research observed that in Chinese individuals, this specific polymorphism is not very common, while in European black race patients with IPF, it is entirely absent, which demonstrates that its occurrence is distinct in different target populations [22].

Simultaneously, there are correlations between MUC5B gene polymorphism and the specific imaging of IPF, as this element is related to the existence and predominance of subpleural fibrosis [45]. Several other factors are described to be involved in developing high values of MUC5B in IPF, such as sequence-binding DNA factors, inflammatory agents, or cellular communication pathways [42].

As of yet, the distinct mechanism through which MUC5B is involved in the development of IPF has not been specified. Many opinions tend to converge toward a debate regarding a possible susceptibility to disease based on MUC5B variants in human lungs. The idea, according to the fact that MUC5B overexpression alters not only mucociliary defense but also the complex process of airway epithelial cell repair, becomes increasingly outlined. Because a significant number of transcriptional factors and inflammatory mediators influence and interfere with MUC5B overexpression, many aspects related to their actions remain to be elucidated, when considering MUC5B intervention [42].

In IPF, MUC5B seems to have predictive value and can also be used for prognosis [43]. The early significance of MUC5B presence in the distal airways may indicate the onset of IPF, guiding efforts toward preventive measures for disease progression [46].

Nevertheless, the existence of MUC5B rs35705950 had no concrete effect on the treatment outcome with nintedanib or pirfenidone, but it suggests a higher survival rate in patients affected by IPF [47,48].

### 3.5. Oncomarkers (CA 15-3, CA 19-9, CA 125, and CEA)

Up to this point, research has highlighted that IPF and lung cancer are similar in some particular aspects, from genetic and epigenetic markers, to risk factors (age, smoking, and work-related or environmental exposure) and several disturbances that occur in the molecules and cells involved in cellular communication pathways [13,49].

A study developed in China showed that, in patients with IPF, the incidence of developing lung cancer is higher than in the general population (23.1%) [49].

Even though tumor markers are usually assessed in cancers, their values are relevant in some benign pathologies, such as interstitial lung diseases or idiopathic lung fibrosis [49]. In IPF, oncomarkers are correlated with pulmonary function, the progression of the illness, and a heightened risk of mortality [50].

The most specific tumor markers used in the management of patients with IPF are carbohydrate antigen (CA) 15-3, CA 19-9, CA 125, and CEA [13,22].

#### 3.5.1. CA 15-3

The MUC1 gene produces CA 15-3, a substantial glycosylated molecule that contains a wide extracellular portion, a membrane-penetrating segment, and a cytoplasmic section, which is well-known as one of the most used markers in breast cancer [13,51].

At the same time, CA 15-3 is relevant as a prognostic marker in IPF, quantifying the severity of the disease, and can be used as an alternative to KL-6 due to its accessibility and lower cost [52,53,54].

A study developed by Rusanov et al. concluded that this biomarker might also be helpful in follow-up since the values of CA 15-3 decrease after pulmonary transplant in individuals with IPF [55].

Nevertheless, there are no notable distinctions found in the connection between CA 15-3 levels and pulmonary function tests or in the survival rates between IPF individuals treated with nintedanib and those who underwent therapy with pirfenidone [52].

#### 3.5.2. CA 19-9

Since Koprowski’s first description of CA 19-9 in 1979 as a carbohydrate produced by the exocrine epithelial cells, this biomarker has been used for the management of gastrointestinal cancers (pancreas, stomach, or colon) [56,57,58]. Nonetheless, increased values of CA 19-9 were detected in several non-malignant conditions, such as IPF or non-tuberculous mycobacterial lung disease [57,58].

On the one hand, higher levels of CA 19-9 were found in end-stage individuals affected by IPF, suggesting that its role could be more specific in predicting the severity rather than the evolution of the disease [59]. On the other hand, the initial serum levels of the biomarker make possible the differentiation between patients affected by progressive disease and those with stable conditions, suggesting that CA 19-9 could also be an effective IPF prognostic biomarker [60].

The reasons behind the increase in CA 19-9 are not fully understood. One theory suggests that injured lungs may release an excess of CA 19-9 as epithelial cells regenerate. Interestingly, significantly affected lungs have been associated with reduced levels of CA-19, potentially linked to the impaired ability to regenerate alveolar epithelium in certain patients. Currently, the molecular involvement of elevated CA 19-9 levels in IPF and other ILDs is speculative [59].

#### 3.5.3. CA 125

CA 125, encoded by the homonymous MUC16 gene, is a glycoprotein that constitutes a significant part of mucus, offering protection against pathogens to the surfaces of different organs (such as the lungs, pleura, peritoneum, endometrium, and endocervix) and is primarily linked to ovarian cancer [61,62].

It is currently used as a biomarker that indicates epithelial injury in progressive IPF due to its substantial secretion by the metaplastic epithelium [54]. The PROFILE cohort study showed that increasing values of CA 125 reflect not only the disease progression but also the overall survival rate, suggesting the potential development of CA 125 as a theranostic marker for evaluating the efficacy of antifibrotic drugs [63]. This theory was confirmed with another study, which highlighted that CA 125 served as a predictor for transplant-free survival in individuals who received antifibrotic treatment. However, the cutoff values were higher in the subjects who underwent therapy in contrast to those who did not [64].

#### 3.5.4. CEA

Carcinoembryonic antigen (CEA), first discovered in 1965 by Gold and Freedman, is a glycoprotein originating from the embryonic endodermal epithelium during fetal development that is typically no longer present in the bloodstream after birth. Still, trace amounts may persist in colon tissue [65,66]. Heightened expression of CEA is found in adenocarcinomas, affecting not only the colon but also other organs (including the pancreas, lungs, ovaries, or breasts) [67].

In roughly 50% of IPF individuals, there is an elevated serum CEA level, which is associated with the severity of the disease [68]. Studies demonstrated that patients with IPF have a noteworthy inverse relation between serum CEA levels and lung function [54].

This biomarker could also serve as a potential indicator for predicting the mortality of pulmonary transplant patients with IPF [69]. Even though its role in IPF is not fully known, CEA could become a substantial biomarker for both diagnosis and prognosis in IPF [54].

### 3.6. Clara Cell Secretory Protein (CC16)

Max Clara described the human Clara cells (also known as club/goblet/exocrine cells) for the first time in 1937 as epithelial cells that do not have cilia and are involved in secretion while being most commonly found in the terminal and respiratory bronchioles [70,71,72,73]. Usually, they represent 9% of the overall airway epithelial cell population in human lungs and have a defense role by releasing different agents such as Clara cell secretory protein (CCSP) and a regenerative function while serving as progenitor cells in case of lung injury [13,73,74]. Besides this prominent localization, Clara cells can also be found in other organs such as the kidneys, prostate, or uterus [73,74]. Their specialized function in processing toxins is also underlined by the high content of cytochrome P450 mono-oxygenases and flavin-containing monooxygenases, which make these cells sensitive to external pollutants and easily influenced by pathogens and harmful molecules [74].

Activated in response to alveolar injury, club cells contribute to injury repair. Although the specific role of CC16 in this process is not fully elucidated, several studies suggest that migrating club cells replace damaged alveoli during injury resolution. The heightened blood capillary permeability of alveoli may aid the diffusion of CC16, secreted by these migratory club cells, into circulation. Considering that IPF predominantly impacts the alveolar epithelium, an elevated CC16 level in the serum could be a reasonable expectation [75].

An elevation of the club cells’ proteins has been detected in the bronchoalveolar lavage (BAL) and blood serum of individuals suffering from different lung diseases, such as sarcoidosis or pulmonary fibrosis, as well in patients mechanically ventilated with elevated positive end-expiratory pressure (PEEP) settings [73]. While the precise reasons remain unclear, these findings could arise from the prompt stimulation of club cells, promoting CC16 secretion, particularly during the initial phases of lung impairment [75].

While being analyzed through SDS-PAGE (sodium dodecyl sulfate-polyacrylamide gel electrophoresis), CCSP exhibits a detectable molecular size of 10 kiloDaltons, leading to its designation as CC10 [76]. Nonetheless, since the precise molecular mass observed through spectrometry is 15.840, a more accurate abbreviation for CCSP is considered to be CC16 [76].

More recent research has focused on determining if the serum values of CC16 might help to differentiate between IPF and non-IPF ILD, but the results concluded that limited sensitivity hinders its role as a possible biomarker in this matter [77]. Ivette Buendía-Roldán et al. also highlighted that Clara cells influence the process of lung injury recovery by using a factor-related apoptosis-inducing ligand (TRAIL) to trigger the death of the epithelial cells localized at the distal airways and alveoli level [77]. Even though TRAIL-expressing Clara cells were found in individuals with IPF, the involvement of CC16 has not yet been studied, leaving this matter as an opportunity for future research [77].

### 3.7. Telomere Shortening

Telomeres represent nucleoprotein complexes composed of repeated TTAGGG hexamer sequences, having as their primary function the protection of the structural integrity and functionality of the terminal regions of linear human chromosomes [78,79]. They contribute to enabling the complete replication of chromosomes, controlling the gene expression and the replicative potential of the cells and their transition into senescence [80].

Telomerase is a ribonucleoprotein complex accountable for the extension of telomeres, which is usually active during pregnancy in immature cells that are not differentiated and in a limited number of lymphocytes [81]. This matter is reflected in considering the shortening of the telomeres as a primary characteristic of the aging process, which happens due to the widely recognized end replication challenge or as a consequence of various biological occurrences [82]. Numerous proteins ensure the proper functioning of the telomerase complex, including telomerase reverse transcriptase (TERT) or telomerase RNA component (TERC) [21].

Telomeres trigger either apoptotic or cellular senescence pathways upon reaching a critical length [83]. Despite the general trend of telomere length diminishing with age, there are significant variations in its average value among different species, and this does not consistently align with life expectancy [82]. A representative indicator of telomere length throughout the body is the length of leukocyte telomeres (LTL), which can be detected in the bloodstream [78].

In IPF, telomere shortening is reported in 25% of the cases, while it is found in 50% of individuals with familial pulmonary fibrosis (FPF); in both situations, its presence suggests a poor prognosis and heightened morbidity post-transplantation [84]. At the same time, in IPF, telomere shortening is linked on its own with a reduced survival rate [22,85]. Shorter telomeres are considered to be a risk factor for the onset of IPF, and their role is also centered on the predictive value and the potential issues associated with transplantation since multiple hematologic complications were detected in patients with both IPF and telomerase mutation [22,81,84]. Aberrant telomere shortening resulting from mutations in TERT and TERC has been identified in 8–15% of individuals with FPF and in up to 3% of sporadic IPF cases [22]. A swifter diminishing in FVC was observed in the IPF cases with shorter telomeres compared to those with longer telomeres, highlighting a notable interplay between the length of telomeres and the reduction in lung function [86]. Future development of new therapies involving telomerase targeting might lead to promising results [86].

### 3.8. Cleaved Cytokeratin 18 (cCK-18)

cCK-18 is a structural protein localized in the epithelial tissues, including the alveoli, which is cleaved twice by caspases in the process of epithelial cell apoptosis and is detectable with M30 antibodies [87].

Indicators of endoplasmatic reticulum stress and the unfolded protein response (UPR) show elevated levels of lung AEC-II in IPF individuals. The endoplasmic reticulum contributes to preserving homeostasis in the presence of unfolded or incorrectly folded proteins through the UPR. If the UPR is unsuccessful in ensuring homeostasis, apoptosis follows. Studies have shown that the alveolar epithelium of the lungs in IPF individuals exhibits activation of the UPR.

cCK-18, found in lung AECs in patients suffering from IPF, is produced through the activation of the UPR in vitro and exhibits distinct elevation in the serum of IPF individuals compared to both normal subjects and those with other ILDs. This fact implies that cCK-18 might serve as a biomarker for the UPR and AEC apoptosis, holding potential as a monitoring tool for therapies that regulate them in patients with IPF [87].

### 3.9. Alpha-v beta-6 (αvβ6) Integrin

Increased signaling of transforming growth factor-β1 (TGF-β1) facilitates the transition of fibroblasts into myofibroblasts, stimulates collagen gene expression, and leads to the accumulation of fibrous tissue that adversely affects pulmonary function. αv integrins are a group of five heterodimeric transmembrane proteins facilitating the transmission of mechanical force between cells and the extracellular matrix. At the same time, they have been identified as crucial mediators of TGF-β activation in fibrosis [88].

Alpha-v beta-6 (αvβ6) integrin, found only in epithelial cells, acts as an activator of transforming growth factor-β1 (TGF-β1), which represents an underlying mechanism of IPF [89,90]. Since its values are elevated in IPF, αvβ6 integrin seems to be relevant as a prognosis biomarker while assessing the disease progression [91,92]. Increased levels of αvβ6 integrins strongly correlate with an unfavorable prognosis [93].

Although biochemically identified over two decades ago, alpha-v beta-1 (αvβ1) integrin has received limited attention. Its composition, involving α and β subunits found in various heterodimers (5 for αv and 12 for β1), has presented challenges in developing antibodies specific to particular heterodimers or deducing function from gene knockout studies [24].

## 4. Discussion

Interstitial lung diseases encompass a wide range of lung conditions arising from inflammation and fibrosis, that affect the pulmonary tissue, which usually manifests through non-specific symptoms like dyspnea and a slow-developing, persistent dry cough, alongside several imagistic anomalies and restrictive patterns in lung function [10]. Idiopathic pulmonary fibrosis is included in this category of diseases, being a condition with uncertain origin, more frequently identified in men usually aged between 50 and 80 years old, having an approximate median survival of around three years [94].

The dysfunction in alveolar epithelial cells (AECs) seems to play an essential role in pathogenesis and gradual development since the breakdown of specific injury-repair processes of the alveoli ultimately leads to repetitive instances of myofibroblast activation followed by the buildup of the extracellular matrix (ECM) [95]. Therefore, the latest efforts in the management of IPF are focused on unraveling the fundamental pathways and genetic factors involved in the trajectory of the disease, uncovering predictive biomarkers for early tracking of the condition, and also monitoring its evolution and treatment response [94].

Our review focused on the actual involvement of the main biomarkers associated with alveolar epithelial cell dysfunction in the management of IPF cases, such as Krebs von den Lungen 6 (KL-6), surfactant proteins, Mucin 5B (MUC5B), tumor markers (CA 15-3, CA 125, CA 19-9, CEA), Clara cell secretory protein (CC16), telomere shortening, cleaved cytokeratin 18 (cCK-18), alpha-v beta-6 (αvβ6), and alpha-v beta-1 (αvβ1) integrins [13,22,24].

Increased values of KL-6 in the serum are also present in other ILDs, tuberculosis, or malignant diseases (lung or breast cancer) [22,23,96]. A serum KL-6 threshold of 500U/mL has been applied to differentiate between patients affected by ILD and subjects who are apparently healthy or suffer from other pulmonary diseases unrelated to ILD [97]. At the same time, some authors consider that the serum KL-6 level might indicate the degree of injury of the epithelium of the alveoli rather than the progression of the disease [98]. Additionally, increased values of KL-6 were correlated with decreased lung function (reduced forced vital capacity (FVC), forced expiratory volume in 1s (FEV1), and total lung capacity (TLC)), which is associated with higher mortality rate [96].

Pulmonary fibrosis and epithelial-mesenchymal transition determine the release of KL-6 as a result of proteolytic cleavage near the plasma membrane, converting it into an effective biomarker not only to diagnose IPF but also to evaluate the severity of the disease or to predict the acute exacerbations and the efficacy of treatment with nintedanib or pirfenidone [25].

Studies have shown a significant increase in SP-A and SP-D serum values in patients with IPF (due to enhanced alveolar vascular permeability), while their levels are diminished in the BAL [99]. At the same time, SP-A and SP-D exhibit different degrees of correlation with the activity and extent of the disease and mortality, which makes them potentially valuable biomarkers [37]. At the same time, high values of SP-D were found in individuals experiencing acute exacerbations of IPF and are also correlated with diminished lung function [13,100].

Simultaneously, surfactant proteins SP-A and SP-D serve as valuable biomarkers for predicting outcomes and tracking therapeutic response since their high levels tend to drop following antifibrotic therapy [38,39].

While the precise significance of Cleaved cytokeratin 18 (cCK-18) and Clara cell secretory protein (CC16) in IPF remains to be fully elucidated, they have the potential to serve as future valuable diagnostic markers.

Some studies have highlighted that when CC16 serum levels are low, an increased risk of mortality is described, particularly in lung cancer [77]. Other studies show that decreased serum values for this protein are present in bronchiolitis obliterans (after allogeneic stem cell transplantation) and lung transplant [74]. At the same time, they correlate with faster deterioration in adult lung function [13]. In contrast, elevated concentrations of CC16 in the BAL and sputum are found in IPF cases [75]. Its serum values are highly increased not only in IPF patients but also in individuals with IPF associated with emphysema; however, they cannot be employed as standalone diagnostic biomarkers [13,75,77].

Some studies showed that in IPF cases, the serum values of cCK-18 are elevated, whereas they remain unaltered in other ILDs, converting this protein into a potentially valuable diagnostic biomarker [22,87]. Nevertheless, the severity or prognosis of IPF was not linked with the baseline level of cCK-18 [101].

At the same time, oncomarkers might not only help to differentiate between IPF and malignancies but could also play an important role in anticipating the severity and prognosis of the disease [13,22].

While it is widely acknowledged that telomere dysfunction primarily impacts the function of lung alveolar epithelial type 2 (AEC-II) cells and significantly contributes to pulmonary fibrosis, it is essential to note that other types of cells (club and basal) could also be affected by telomere dysfunction, influencing their functions accordingly [102]. Nevertheless, the mechanism by which dysfunctional AEC cells contribute to fibrogenesis remains unclear. One potential explanation could be their impaired signaling to other cells. A study using a transgenic animal model proposed that it was not the apoptosis, but the senescence of AEC-II cells, that promotes the onset of pulmonary fibrosis. A notable discovery from a mouse model highlighted that the selective removal of the cell division control protein 42 homolog (Cdc42) in AEC-II cells results in the gradual development of pulmonary fibrosis. This study uncovered that the lack of Cdc42 blocks the transition of AEC-II into AEC-I. This phenomenon inhibits the formation of new alveoli, thereby contributing to an uneven distribution of mechanical tension in the lungs [102].

Another promising prognostic indicator might be represented by alpha-v beta-6 (αvβ6) integrin since its elevated levels suggest an unfavorable evolution [93]. Decaris et al. conducted a study where the effect of inhibiting αvβ6 and αvβ1 individually and simultaneously was compared, revealing a distinct additive result. Blocking both integrins demonstrated a decreased expression of collagen in ex vivo cultures of pulmonary tissue cultures from IPF individuals. Moreover, by not disrupting systemic TGF-β activity, dual αvβ6 and αvβ1 inhibition may lower the risk of toxicities associated with the complete inhibition of the TGF-β pathway. A therapeutic strategy blocking both αvβ6 and αvβ1 simultaneously may operate through distinct mechanisms compared to conventional drugs like nintedanib or pirfenidone. This raises the potential for enhanced antifibrotic benefits in IPF patients through the combined use of these drugs [88].

Since surface secretory cells across the respiratory tract and submucosal gland generate MUC5b, its function against pathogens is essential [41]. Its “rs35705950” variant correlates with excessive mucin production and buildup in the distant airspaces, potentially affecting the mechanical defense process, which in turn might initiate the cough reflex [46]. Therefore, there might be a link between the presence of this variant and the intensity of the cough in patients with IPF [45].

When assessing the severity of IPF, MUC5B could prove to be a highly valuable indicator [45]. Considering the genetic factors involved in developing IPF, investigating the impact of telomere shortening in its pathogenesis could also offer significant insights, as it is associated with a poor prognosis and increased post-transplant morbidity [84].

Although there has been significant advancement in the evaluation of these biomarkers, indicating their potential involvement in enhancing the ease of diagnosis, improving prognostic accuracy, and leading to the discovery of new therapeutic approaches, their global and standardized utilization remains limited due to a lack of comprehensive understanding of their implications. Investigation of the other categories of relevant biomarkers in IPF (those associated with extracellular matrix remodeling, fibrogenesis, and fibroproliferation and biomarkers related to immune dysfunction and inflammation) and their relationship with biomarkers associated with alveolar epithelial cell damage and dysfunction could represent an essential avenue for research and hold significant potential for advancing our understanding of this disease.

This study is limited by the fact that there is an abundance of approaches regarding the need of systematizing and finding the “silver bullet” for the prevention, early diagnosis, and early treatment of IPF. However, the lack of specificity of these biomarkers and the diversity and heterogeneity of ILDs places the use of biomarkers in the diagnostic process as only a possible link in a long chain.

## 5. Conclusions

There is evident research progress in molecular biomarkers involved in or related to idiopathic pulmonary fibrosis. Their clinical validation has to be correlated with an improvement in IPF management and with new therapeutic strategies. Further consistent studies are necessary to explore, improve, and define the complex role and efficacy of these biomarkers in predicting the diagnosis, prognosis, and therapeutic response of IPF.

## Figures and Tables

**Figure 1 jpm-13-01607-f001:**
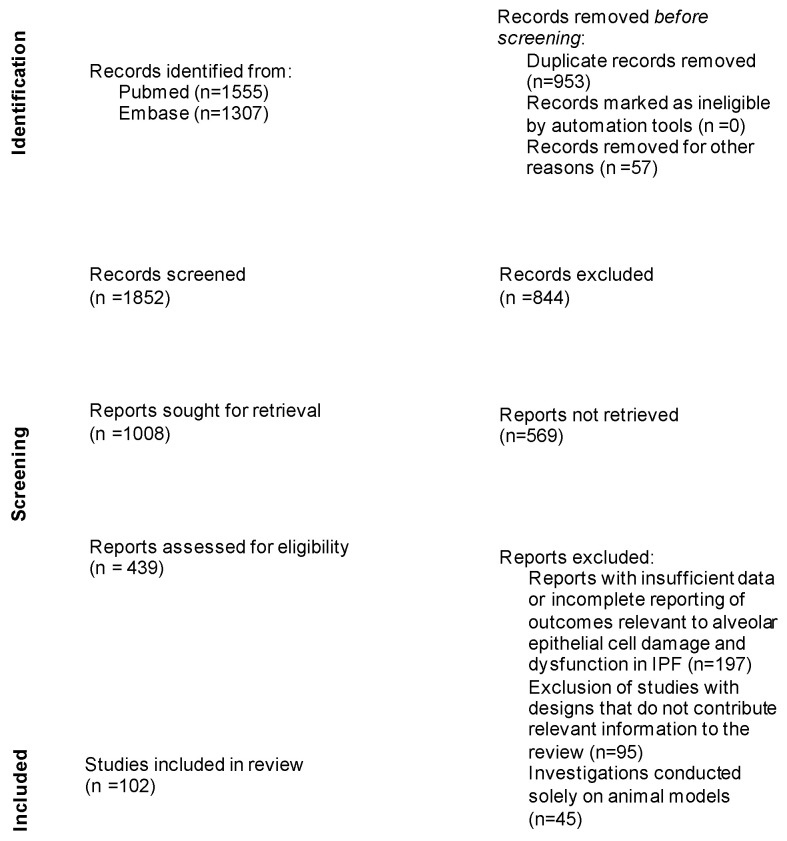
Prisma flow diagram illustrating the article selection process.

## Data Availability

All information provided in this review is supported by the relevant references.

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
