# Peer review of "Exploring the Role of Biomarkers Associated with Alveolar Damage and Dysfunction in Idiopathic Pulmonary Fibrosis—A Systematic Review"

_jpm, 2023, doi:10.3390/jpm13111607_

Round 1

Reviewer 1 Report

Comments and Suggestions for Authors

I the context with Idiopathic Pulmonary Fibrosis (IPF), this review article focusses on biomarkers associated with alveolar epithelial cell damage and dysfunction. Eight different classes are extensively described. Some more major and minor comments can be found below.

The following sentence is contained in the text: ‘An ideal biomarker should be easily obtainable, consistently measurable and suitable for repeated monitoring, while offering several benefits compared to current medical assessments in terms of simplicity, time efficiency or cost-effectiveness.’

Thus, in the Discussion section, the eight classes should be addressed and critically discussed in the order of appearance.

Major comments

Title: This reviewer does neither see any tip of the iceberg nor does he understand ‘honest’ in that context. I.e. the phrasing seems somewhat blatantly.

Abstract; Conclusion: This sentence does not make much sense. If further studies are needed, then do them. Please, have in mind that so many other studies are needed.

Use of English language should be improved. L. 99+100: Is the following meant? The disruption of the basement membrane in alveolar epithelial cells is a crucial characteristic of idiopathic pulmonary fibrosis (IPF), leading to abnormal activation of these cells.

l. 75-76: Search terms need to be mentioned

l. 76-77: Which criteria were used?

Materials and Methods: Number of references screened needs to be described. Also, number of references, finally included in the text.

Minor comments

l. 1: Should you decide on the type?

l. 46: what do you mean by: …. and pulmonary fibrosis, classified based on several histopathological

l. 80: Within a review article, own experiments are of no use.

l. 97: what does BAL mean?

Comments on the Quality of English Language

Parts of the English could be improved, such as use of articles.

Reviewer 2 Report

Comments and Suggestions for Authors

·      This review gives a description on the various biomarkers of alveolar epithelial cell damage and dysfunction.  Most of the contents are descriptive, lacking mechanism manifestation at molecular level. Authors should clearly state their own mechanism separately.

·      The review is not comprehensive. Besides the biomarkers mentioned in the review, there is omission of other biomarkers. These should be discussed.

·      The title of this article is redundant with other review articles on biomarkers of alveolar epithelial cell damage and dysfunction.  The article rehashes other views and conclusions from some primary literature, without critique or insight.

Comments on the Quality of English Language

On the whole, the manuscript is well written, but some vocabularies should be revised, I suggest authors to carefully revise all throughout the text to use the proper term. And, if possible, the manuscript may be reviewed and corrected by a native English speaker.
